# CoAct-1: Computer-using Multi-agent System with Coding Actions

**Linxin Song**[1,*], **Yutong Dai**[2], **Viraj Prabhu**[2], **Jieyu Zhang**[3], **Taiwei Shi**[1], **Li Li**[1], **Junnan Li**[2]
**Silvio Savarese**[2], **Zeyuan Chen**[2], **Jieyu Zhao**[1], **Ran Xu**[2], **Caiming Xiong**[2]
[1]University of Southern California  [2]Salesforce  [3]University of Washington

Code: https://github.com/SalesforceAIResearch/CoAct-1

## Abstract

Autonomous agents that operate computers via Graphical User Interfaces (GUIs) often struggle with efficiency and reliability on complex, long-horizon tasks. While augmenting these agents with planners can improve task decomposition, they remain constrained by the inherent limitations of performing all actions through GUI manipulation, leading to brittleness and inefficiency. In this work, we introduce a more robust and flexible paradigm: enabling agents to use coding as an enhanced action. We present CoAct-1, a novel multi-agent system that synergistically combines GUI-based control with direct programmatic execution. CoAct-1 features an Orchestrator that dynamically delegates subtasks to either a conventional GUI Operator or a specialized Programmer agent, which can write and execute Python or Bash scripts. This hybrid approach allows the agent to bypass inefficient GUI action sequences for tasks like file management and data processing, while still utilizing visual interaction when necessary. We evaluate our system on the challenging OSWorld and WindowsAgentArena benchmark, where CoAct-1 achieves a new state-of-the-art success rate of 60.8% on OSWorld and 52.5% on WindowsAgentArena, significantly outperforming prior methods. Furthermore, our approach dramatically improves efficiency, reducing the average number of steps required to complete a task to just 10.15 on OSWorld, compared to 15 for leading GUI agents. Our results demonstrate that integrating coding as a core action provides a more powerful, efficient, and scalable path toward generalized computer automation.

## 1 Introduction

Recent advancements in computer-using agents have primarily focused on operating through Graphical User Interfaces (GUIs). While these GUI agents, powered by vision-language (action) models (Li et al., 2023; Deepmind, 2025a;b; OpenAI, 2024; 2025a; Qin et al., 2025; Xie et al., 2025a; Yang et al., 2025), have demonstrated the ability to perform a variety of tasks, they often struggle with long-horizon planning and interactions in environments with dense GUI elements. For example, routine tasks in Office productivity software often involve a long and intricate sequence of precise GUI operations, such as locating a specific table within a multi-sheet spreadsheet, filtering it based on complex criteria, copying the results, and saving them as a new CSV file. Similarly, tasks like finding all image files in a nested directory structure, resizing them to specific dimensions, and then compressing the entire directory into a single archive are brittle and inefficient when solved via GUI actions such as clicking and dragging. In these scenarios, existing agents often struggle with visual grounding ambiguity (e.g., mistaking visually similar icons or menu items) and the cumulative probability of errors over long-term interactions. A single mis-click or misunderstood UI element can derail the entire task.

To address these challenges, a prominent line of research has focused on augmenting GUI agents with dedicated high-level planners. Approaches such as GTA-1 (Yang et al., 2025) and other modular

---

*Work done during the internship at Salesforce.

systems (Yang et al., 2024; Xu et al., 2024; Agashe et al., 2024; 2025) utilize powerful language models like OpenAI o3 (OpenAI, 2025b) to decompose a user's high-level goal into a sequence of more manageable subtasks. This hierarchical decomposition can improve performance on complex, multi-step problems by providing a structured plan. However, this paradigm does not fundamentally address the inefficiency and brittleness associated with exclusive reliance on GUI-based execution. Even with the high-level planning, the agent still needs to navigate menus, click buttons, and type into fields, even for operations that could be accomplished more directly and reliably through programmatic means. This leaves the system susceptible to planning uncertainty, visual perception errors, and the integration challenges between high-level planning and low-level action generation.

In this work, we advocate for and instantiate a more flexible and powerful action space. We propose a hybrid approach that combines the intuitive, human-like strengths of GUI manipulation with the precision, reliability, and efficiency of direct system interaction through code. We introduce **CoAct-1** (Computer-using Multi-agent System with **Co**ding **Act**ions), a novel multi-agent system composed of three specialized agents: Orchestrator, Programmer, and GUI Operator. A high-level Orchestrator serves as the central planner, decomposing the user's goal and determining the appropriate modality for each subtask. Based on this analysis, it assigns the task to one of two distinct execution agents: a Programmer agent, which writes and executes Python or Bash scripts for backend operations like file management, data processing, or environment configuration; or a GUI Operator, a VLM-based agent that performs frontend actions like clicking buttons and navigating visual interfaces. This dynamic delegation allows CoAct-1 to strategically bypass inefficient GUI sequences in favor of robust, single-shot code execution when appropriate, while still leveraging visual interaction for tasks.

Our experimental analysis provides strong evidence for the advantages of this hybrid design. On the OSWorld and WindowsAgentArena benchmark, CoAct-1 establishes a new state-of-the-art, achieving an overall success rate of 60.76% and 52.50%, respectively. This marks a significant improvement over leading baselines like Agent S2.5 (55.98%) on OSWorld. The performance gains are most pronounced in categories where programmatic control is highly advantageous. For instance, in Calc (70.21%), multi-application (47.88%), and VS Code (78.26%) tasks, our Programmer's ability in executing precise scripts leads to substantial gains over the strongest GUI-only methods. Beyond improving success rates, our dual-modality approach dramatically enhances operational efficiency. By replacing long, error-prone click sequences with concise code, CoAct-1 solves tasks in an average of just 10.15 steps on OSWorld, a stark contrast to the 15 steps required by agents like GTA-1. This efficiency underscores the potential of our approach to pave a more robust and scalable path toward generalized computer automation.

## 2 RELATED WORK

**Screen parsing and visual grounding** A first line of work focuses on perceiving and grounding GUI elements directly from pixels, without relying on DOM or accessibility hooks. *OmniParser* learns screen-parsing primitives for pure vision–based understanding (Lu et al., 2024). On the grounding side, *SeeClick* (instruction-to-target grounding), *Aria-UI* (instruction grounding over GUIs), and *UGround* (universal GUI grounding) map language to actionable screen locations (Cheng et al., 2024; Yang et al., 2024; Gou et al., 2024). *OS-Atlas* trains a *foundation action model* to generalize across diverse interfaces (Wu et al., 2024). Dedicated grounding evaluations such as *ScreenSpot-Pro* further benchmark grounding under professional, high-resolution settings (Li et al., 2025).

**Native end-to-end GUI agents** A third thread trains *native* agents that unify perception, reasoning, and action in a single model. *UI-TARS* and *OpenCUA* exemplifies this approach with a unified action space for mouse/keyboard operations across apps, eschewing hand-crafted controllers (Qin et al., 2025; Wang et al., 2025a;b). *AGUVIS* pushes toward unified, pure-vision GUI agents that generalize across interfaces (Xu et al., 2024).

**Modular planner–grounder agents** A second strand explicitly separates *what to do* from *where/how to act on screen*: a language planner proposes subgoals while a visual model grounds each step. Representative systems include *SeeClick* and *OS-Atlas* (Cheng et al., 2024; Wu et al., 2024). *GTA-1* strengthens this two-stage paradigm via *test-time scaling*: sampling multiple candidate actions and using an MLLM judge to select among them, improving robustness on high-resolution, cluttered UIs (Yang et al., 2025). Other related open frameworks such as *Agent-S / Agent-S2* and *AutoGen*

provide reusable infrastructures for multi-agent orchestration and tool calling (Agashe et al., 2024; 2025; Song et al., 2025a; Zhang et al., 2024; Wu et al., 2023; Zhang et al., 2023; 2025b).

**Hybrid agentic frameworks** Beyond GUI-only interaction, several agentic systems compose tools and APIs on the fly to extend capabilities at run time. Examples include *UFO-2* (Zhang et al., 2025a), *PyVision* (Zhao et al., 2025), *BeyondBrowsing* (Song et al., 2025b) and *ALITA* (Qiu et al., 2025), which, while not restricted to GUI/CUA, share the principle of dynamically constructing and invoking tools.

# 3 COMPUTER-USING AGENT WITH CODING AS ACTIONS

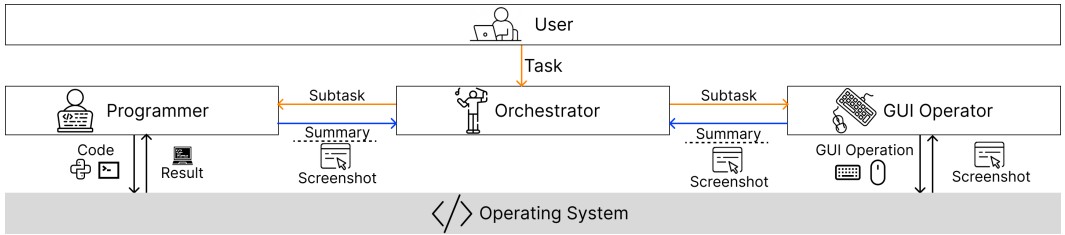

**Figure 1:** Multi-agent system design for our CoAct-1. This multi-agent system includes a Programmer that can interact with the operating system through multi-round coding. This multi-agent system includes an **Orchestrator**, which serves as the high-level planner that decomposes goals and delegates subtasks to the appropriate execution agent, a **Programmer**, which interacts with the operating system through multi-round coding, and a **GUI Operator**, which leverages vision-language capabilities to perform visual interface actions.

In this work, we introduce a new system-interactive action: coding, to replace part of the redundant and brittle GUI actions. Unlike summarizing APIs or SDKs from each application or website, we focus on enabling agents to perform free-form coding to solve computer-use problems guided by a strong language model. Specifically, we design a multi-agent system, CoAct-1, that introduces a new agent, Programmer, capable of interacting with the OS through a coding-observation circle. An Orchestrator serves as the high-level controller, determining whether to assign the subtask to the Programmer or the GUI Operator. The overall framework is illustrated in Figure 1.

## 3.1 PROBLEM DEFINITION

We formalize the problem of general computer control as an interactive decision-making process. At each timestep $t$, the agent observes the computer environment (primarily consists of a screenshot) $o_t \in \mathcal{O}$, and takes an action $a_t \in \mathcal{A}$ according to a policy $\pi(a_t | H_t, G)$. Here, $H_t = (o_1, a_1, ..., o_{t-1}, a_{t-1}, o_t)$ represents the historical context, and $G$ is the user's high-level goal provided in natural language. Learning an effective policy is particularly challenging when the action space $\mathcal{A}$ is restricted to low-level GUI operations. Complex tasks, such as managing nested files or processing spreadsheet data, can require long and intricate sequences of GUI actions. This makes the process inefficient and highly susceptible to error propagation, where a single mis-check can derail the entire task.

To address this limitation, we introduce a hybrid action space that integrates direct programmatic control. We augment it to the standard GUI action space, denoted as $\mathcal{A} = \mathcal{A}_{\text{GUI}} \cup \mathcal{A}_{\text{Code}}$. An action $a_t \in \mathcal{A}_{\text{GUI}}$ involves the direct manipulation of the graphical interface (e.g., mouse clicks, keyboard typing). In contrast, an action $a_t \in \mathcal{A}_{\text{Code}}$ consists of a Python or Bash script that interacts directly with the operating system's backend. This allows the agent to perform complex operations like file manipulation or data processing in a single, robust step, effectively bypassing brittle and inefficient GUI sequences. In CoAct-1, the policy $\pi$ is implemented hierarchically. A high-level Orchestrator acts as a meta-policy, $\pi_{\text{orch}}$, which analyzes the current subtask and delegates it to one of two specialized executor policies: a GUI Operator that implements $\pi_{\text{GUI}}$ for actions in $\mathcal{A}_{\text{GUI}}$, or a Programmer that implements $\pi_{\text{Code}}$ for actions in $\mathcal{A}_{\text{Code}}$.

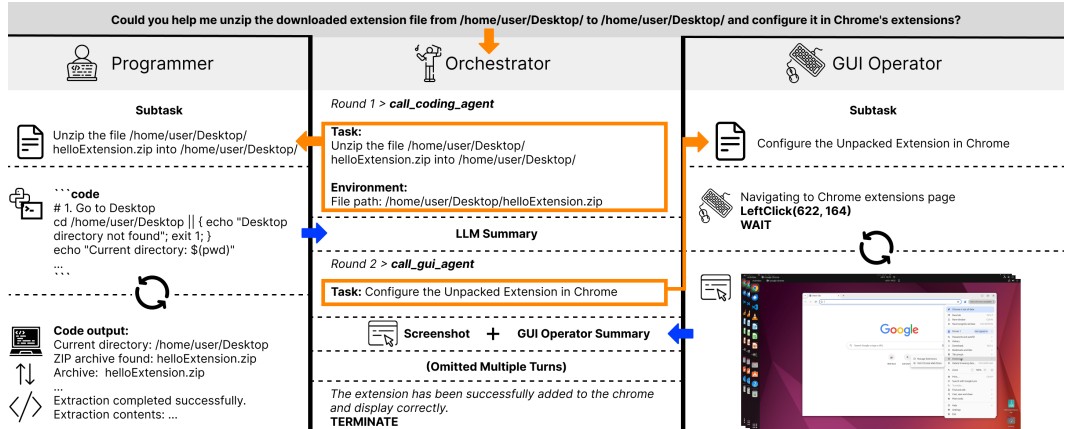

**Figure 2:** Illustration of CoAct-1 workflow. Given a user task, Orchestrator can choose either to call Programmer or GUI Operator to solve a subtask. A programmer can interact with the OS by coding, and a GUI Operator can interact with the OS by performing GUI operations.

## 3.2 MULTI-AGENT SYSTEM DESIGN FOR COMPUTER USE

Our multi-agent system is the architectural instantiation of the hierarchical policy $\pi$ outlined in the problem definition. It comprises three specialized agents—the Orchestrator, Programmer, and GUI Operator that collaboratively generate the action trajectory $\tau$ to solve the user's goal $G$. Each agent establishes a dedicated conversation to perform its role.

**Orchestrator.** The Orchestrator embodies the high-level meta-policy, $\pi_{\text{Orch}}$. It is responsible for task decomposition and dynamic planning based on the full history of observations $H_t$ and the overall goal $G$. The Orchestrator does not interact directly with the OS. Instead, its primary function is to select the best specialized sub-policy, $\pi_{\text{Code}}$ or $\pi_{\text{GUI}}$, to execute the current subtask. Upon completion, the Orchestrator receives a summary of the execution process and a new observation $o_{t+1}$ (a screenshot along with a summary reflecting the current system state) to inform its next decision. If it determines that the overall goal $G$ is met, it outputs a termination signal.

**Programmer.** The Programmer implements the specialized policy $\pi_{\text{Code}}$, responsible for generating actions $a_t \in \mathcal{A}_{\text{Code}}$. Upon receiving a subtask from the Orchestrator, it initiates a multi-round conversation with a code interpreter. It generates Python or Bash scripts, which are executed by the interpreter. The feedback, consisting of the code execution results, allows the Programmer to reflect and refine their code until the subtask is solved. The Orchestrator provides the Programmer with sufficient context, such as file paths or window information inferred from $H_t$, to ground its code generation.

**GUI operator.** The GUI agent is a vision-language action model that implements the GUI-based policy $\pi_{\text{GUI}}$ for generating actions $a_t \in \mathcal{A}_{\text{GUI}}$. Similar to the Programmer, the GUI Operator engages in a multi-round interactive loop to complete its assigned subtasks. In each step of this "perception-action" loop, the agent takes the current screenshot and the subtask instruction as input to generate a single GUI action (e.g., a mouse click or keyboard input). A GUI action interpreter executes this action on the OS, which in turn provides a new screenshot as observational feedback. The GUI Operator uses this new visual information to decide on its subsequent action, continuing this cycle until its subtask is complete.

## 3.3 WORKFLOW AND MEMORY DESIGN

The hierarchical policy implemented by CoAct-1 necessitates a structured workflow and memory system to manage the flow of information between agents. This design ensures that each agent has the necessary context without being overwhelmed by irrelevant details from other parts of the task. The overall workflow is illustrated in Figure 2.

**Workflow** The workflow begins when the Orchestrator, acting as the meta-policy $\pi_{\text{Orch}}$, delegates a subtask to an appropriate executor agent: either the Programmer or the GUI Operator. The selected agent then engages in its own multi-round interactive loop to solve the subtask, generating a detailed conversational history of its actions and the environment's responses. Upon completing the subtask, this detailed history is processed by a dedicated summarizer model. This model condenses the entire interaction into a concise summary that captures the key actions taken and the final outcome. This summary, along with the final screenshot representing the new state of the environment, is then returned to the Orchestrator. This handoff mechanism provides the Orchestrator with a condensed, high-level update to its overall task history $H_t$.

**Memory design for task decomposition** CoAct-1 employs a hierarchical and isolated memory structure:

- **Orchestrator Memory**: The Orchestrator maintains the long-term, primary memory, which corresponds to the historical context $H_t$ from our problem definition. It consists of the initial user goal $G$ and the sequence of summaries and screenshots received from the executor agents after each completed subtask. This aggregated history provides the context for all high-level planning decisions.
- **Executor Memory**: The Programmer and GUI Operator each maintain a short-term, working memory that is active only for the duration of their assigned subtask. This memory contains the "instance conversation history" of their multi-round interaction with the OS.

To ensure modularity and focus, these memories are isolated; the agents do not share their conversational histories directly. Furthermore, once an executor completes its subtask and reports back to the Orchestrator, its working memory is cleared. This reset mechanism is critical, as it allows the executor agents to focus entirely on the context of the new subtask they receive without being influenced by prior, irrelevant interactions.

## 4 EXPERIMENTS

### 4.1 BENCHMARK DATASETS

We evaluate CoAct-1 on OSWorld (Xie et al., 2024) and WindowsAgentArena (Bonatti et al., 2024). Both are scalable real-computer testbed that exposes an OS (Windows or Ubuntu) to an agent through pixel streams and an OS shell interface. OSWorld comprises 369 tasks, while WindowsAgentArena comprises 154 tasks. These task span common productivity tools, IDEs, browsers, file managers, and multi-application workflows, thereby challenging both vision–language grounding and long-horizon planning in heterogeneous GUI environments.

### 4.2 BASELINES

We compare CoAct-1 with two categories of computer-using agents, the end-to-end models and agentic methods. These baselines represent the forefront of GUI-based task automation.

**End-to-end model** An end-to-end model takes user instructions and OS screenshots as input and outputs corresponding actions by pure inference without any agentic workflow. *OpenAI o3* (OpenAI, 2025b) is a cutting-edge reasoning model from OpenAI that excels at multi-step problem-solving and versatile, context-aware assistance. *OpenAI CUA 4o* (OpenAI, 2025a) uses vision and reasoning to interact with graphical user interfaces, controlling the mouse and keyboard to perform tasks. It is the technology behind services like Operator and ChatGPT agent. *UI-TARS* (Qin et al., 2025) UI-TARS introduces a fully end-to-end, screenshot-only native GUI agent model that unifies perception, reasoning, memory, and action. *OpenCUA(32B)* (Wang et al., 2025b) introduces a fully open-source framework, including a scalable data collection tool, the first large-scale multi-OS computer-use task dataset, a reflective chain-of-thought reasoning pipeline, and strong vision-language agent models.

**Agentic Method** The agentic method encompasses single- and multi-agent systems with diverse structures, such as planner-grounder, planner-multigrounder, etc. These baselines primarily focus on enhancing the ground-level ability of a language model to improve computer-based performance.

*Jedi-7B w/ o3 (Xie et al., 2025b)* We refer the term Jedi to a Qwen2.5-VL trained on the Jedi dataset. The author plugs Jedi into an agent stack to translate high-level plans into pixel-perfect GUI actions, achieving large gains on OSWorld, WindowsAgentArena, and multiple grounding benchmarks. *GTA-1 w/ o3 (Yang et al., 2025)* The GUI Test-time Scaling Agent (GTA-1) is a GUI agent that addresses the challenges of planning ambiguity and action grounding in high-resolution interfaces. To improve accuracy, this agent employs a "test-time scaling" strategy, where it generates multiple possible actions and utilizes a Judge model to select the optimal one. It also leverages the GRPO to train a powerful grounder. *Agent S2.5 w/ o3 (Agashe et al., 2025)* Agent S2.5 is a compositional planner-multigrounder framework in which a planner generates high-level subgoals, multiple grounders executes them while delegating GUI-element localization to visual, textual, and structural experts via a Mixture-of-Grounding, and both levels proactively replan after every subgoal to remain robust to changing screens.

Besides the above powerful baselines, we also add Agent S (Agashe et al., 2024) and NAVI (Bonatti et al., 2024) as baselines for WindowsAgentArena.

## 4.3 IMPLEMENTATION DETAILS

**Environment** We test the CoAct-1 on Linux with an extended RESTful server from OSWorld. Specifically, we implement a remote code interpreter that can take long Python and Bash scripts as input and return the execution result back to the sender. On the other hand, for each task, OSWorld will establish an initial state, such as opening a set of apps or specific websites, or downloading the specified files to a specified location, etc. After the initial state is ready, we will take a screenshot as the initial input along with a user task to CoAct-1 and baselines.

**CoAct-1 settings** We implement CoAct-1 using AG2 (Wu et al., 2023). In CoAct-1, we adopt OpenAI o3 for Orchestrator and OpenAI o4-mini for Programmer. For the GUI Operator, we use OpenAI computer-use-preview, a vision-language action model finetuned by OpenAI for computer use, as the backbone model. We use the o4-mini as the summarizer for summarizing the conversation history between the Programmer and the Orchestrator. We set the maximum round $I$ for the Programmer to 20, the maximum step $K$ for the GUI Operator to 25, and the maximum round $J$ for the Orchestrator to 15. Therefore, the number of system interactions, i.e, the number of steps, for CoAct-1 is upper bounded to 375 (but in all cases, as shown in Figure 3d, CoAct-1 will early stop before 150 steps). More details are in Appendix C.

**Evaluation Protocol** We evaluate our method with the rule-based evaluator provided by OSWorld and WindowsAgentArena. Internally, every evaluator is expressed as a Boolean expression built from 134 atomic, execution-based evaluators that the authors handcrafted for the benchmark. For a given task, the benchmark composes these atoms with logical AND / OR operators, so a "pass" might require, for instance, (file exported AND MD5 matches) AND (email sent == True).

## 4.4 RESULTS

Our experimental results, detailed in Table 1 and Table 2, unequivocally establish CoAct-1 as the new state-of-the-art across two challenging, real-world computer operation benchmarks. The findings validate our core hypothesis: integrating programmatic actions alongside traditional GUI manipulation provides a more robust, efficient, and generalizable paradigm for computer automation.

**Performance on OSWorld** On the comprehensive OSWorld benchmark (Table 1), CoAct-1 demonstrates superior performance and efficiency. It achieves a final success rate of 60.76% within the 150-step limit, creating a significant margin over the strongest contemporary agentic frameworks, including Agent S2.5 w/ o3 (55.98%) and GTA-1 w/ o3 (53.07%). The strength of our hybrid architecture is not only in its peak performance but also in its consistency across different task complexities. At the 100-step mark, CoAct-1 already leads with a 59.93% success rate, surpassing the final scores of all other baselines. The advantages of our approach are particularly pronounced in categories where programmatic control is most effective. CoAct-1 achieves top-tier performance in tasks requiring complex data or file system interactions, scoring 64.80% in Office, 75.00% in OS, and 47.87% in Multiple Apps. This exceptional performance in domains that are historically brittle for pure-GUI agents underscores the efficacy of delegating backend operations to the Programmer agent,

**Table 1:** Comparison of the state-of-the-art methods on the OSWorld (Xie et al., 2024) verified benchmark. We split the results by steps and show the approach type in the second column. **Office** tasks include tasks from LibreOffice Calc, LibreOffice Impress, and LibreOffice Writer. **Daily** tasks include tasks from Chrome, Thunderbird, and VLC. **Professional** includes tasks from GIMP and VSCode. We report the success rate (%) as the evaluation metric for each type of task, and mark the best result of each step budget in **bold** and the best result overall step budgets in **underlined bold**.

| Agent Model | Office (104 tasks) | Daily (78 tasks) | Professional (48 tasks) | OS (24 tasks) | Multiple Apps (101 tasks) | Avg. |
|---|---|---|---|---|---|---|
| | | | *15 steps* | | | |
| OpenAI o3 | 1.45 | 8.02 | 12.29 | 37.50 | 11.82 | 9.09 |
| UI-TARS-1.5 (7B) | 27.19 | 27.99 | 61.45 | 34.78 | 5.38 | 25.76 |
| OpenAI CUA 4o | 22.17 | 37.65 | 41.22 | 45.83 | 10.75 | 26.01 |
| OpenCUA | 26.69 | 33.25 | 51.57 | 43.48 | 10.41 | 28.12 |
| Agent S2.5 w/ o3 | 42.85 | 44.61 | 57.10 | **70.83** | 17.82 | 38.98 |
| Jedi-7B w/ o3 | 45.84 | **57.49** | **60.95** | 50.00 | 20.43 | **42.37** |
| CoAct-1 | **47.18** | 42.30 | 47.74 | 66.67 | **23.82** | 39.81 |
| | | | *50 steps* | | | |
| OpenAI o3 | 11.50 | 19.78 | 30.10 | 37.50 | 11.82 | 17.17 |
| UI-TARS-1.5 (7B) | 26.51 | 31.41 | 48.91 | 25.00 | 9.77 | 25.08 |
| OpenAI CUA 4o | 23.56 | 38.43 | 52.09 | 70.83 | 15.86 | 31.19 |
| OpenCUA | 30.06 | 42.31 | 58.28 | 47.83 | 16.79 | 33.76 |
| GTA-1-7B w/ o3 | 48.58 | 52.31 | **77.84** | 58.33 | 37.05 | 48.59 |
| Jedi-7B w/ o3 | 50.10 | **65.25** | 68.65 | 54.17 | 34.97 | 50.65 |
| Agent S2.5 w/ o3 | 52.81 | 55.80 | 75.42 | **75.00** | 39.53 | 54.21 |
| CoAct-1 | **62.91** | 59.43 | 69.89 | 70.83 | **42.37** | **56.38** |
| | | | *100 steps* | | | |
| OpenAI o3 | 17.23 | 26.29 | 38.79 | 62.50 | 16.53 | 23.00 |
| UI-TARS-1.5 (7B) | 25.01 | 31.07 | 46.99 | 29.17 | 8.80 | 25.41 |
| OpenAI CUA 4o | 25.04 | 39.19 | 55.43 | 58.33 | 18.48 | 31.38 |
| OpenCUA | 30.06 | 38.89 | 60.70 | 52.17 | 18.10 | 33.84 |
| Jedi-7B w/ o3 | 47.89 | 64.37 | **75.92** | 50.00 | 35.27 | 50.98 |
| GTA-1-7B w/ o3 | 55.68 | **64.74** | 61.20 | 62.50 | 38.34 | 53.07 |
| Agent S2.5 w/ o3 | 54.23 | 55.80 | 75.42 | **75.00** | 44.06 | 55.98 |
| CoAct-1 | **64.80** | 61.60 | 71.82 | **75.00** | **47.87** | 59.93 |
| | | | *150 steps* | | | |
| CoAct-1 | **64.80** | **66.51** | 71.82 | **75.00** | **47.87** | **60.76** |

**Table 2:** Comparison of the state-of-the-art methods on the WindowsAgentArena (Bonatti et al., 2024). We split the results by steps and show the approach type in the second column. **Office** tasks include tasks from LibreOffice Calc and LibreOffice Writer. **Web** tasks include tasks from Chrome, and Microsoft Edge. **Windows System** includes tasks from settings and File Explorer. **Windows System** includes tasks from Settings and File Explorer. **Windows Utils** includes tasks from Clock, Windows Calculator, Notepad, and Microsoft Paint. We report the success rate (%) as the evaluation metric for each type of task and mark the best result in **bold**.

| Method | Office(43 tasks) | Web(30 tasks) | Windows System(24 tasks) | VSCode(24 tasks) | VLC(21 tasks) | Windows Utils(12 tasks) | Avg. |
|---|---|---|---|---|---|---|---|
| Agent S | 0.0 | 13.3 | 45.8 | 29.2 | 19.1 | 22.2 | 18.2 |
| NAVI | 0.0 | 27.3 | 33.3 | 27.3 | 30.3 | 8.3 | 19.5 |
| Agent S2 | 7.0 | 16.4 | 54.2 | **62.5** | 28.6 | 33.3 | 29.8 |
| CoAct-1 (15 steps) | 8.7 | 3.3 | 50.0 | 29.2 | 23.8 | 44.4 | 21.4 |
| CoAct-1 (50 steps) | 26.1 | 33.3 | 75.0 | 54.2 | 42.4 | 55.6 | 43.5 |
| CoAct-1 (100 steps) | **30.4** | **50.0** | **83.3** | **62.5** | **47.2** | **77.7** | **52.5** |

which can execute precise and reliable scripts for tasks involving spreadsheets, file manipulation, and cross-application data flows.

**Performance on WindowsAgentArena** We further evaluated CoAct-1 on the WindowsAgentArena benchmark (Table 2). The results show that our framework successfully transfers its capabilities to a different operating system, again achieving state-of-the-art performance by a substantial margin. CoAct-1 attains an overall success rate of 52.5%, which is a remarkable improvement over prior leading methods like Agent S2 (29.8%) and NAVI (19.5%). The performance breakdown on WindowsAgentArena further reinforces our central claim. CoAct-1 shows commanding strength in system-level and utility-based tasks, achieving standout scores of 83.3% in Windows System and 77.7% in Windows Utils. These categories, which involve interacting with file explorers, system settings, and other native utilities, are ideally suited for the script-based actions of the Programmer. Furthermore, CoAct-1's performance demonstrates clear and effective scaling with an increased step

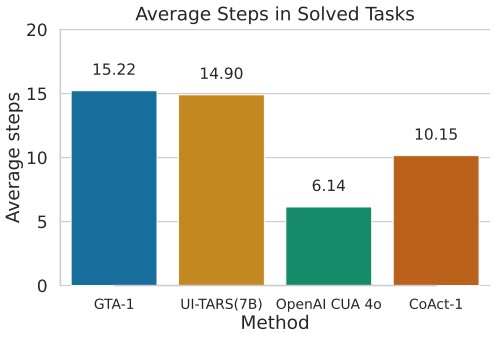

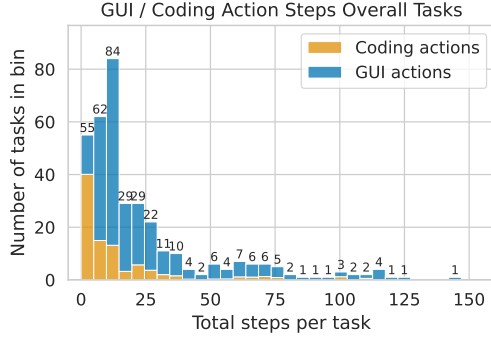

**(a)** Average steps per passed task with 100 step budget, showing CoAct-1 is significantly more efficient than other SOTA agentic frameworks like GTA-1.

**(b)** Distribution of tasks by total step count, illustrating the ratio of coding to GUI actions and showing that coding helps reduce the total action steps.

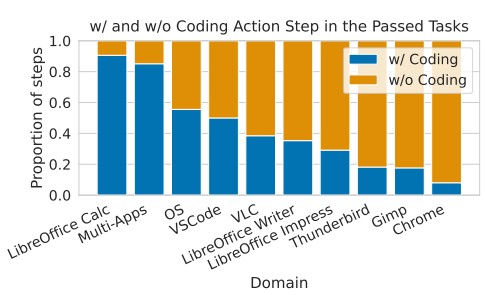

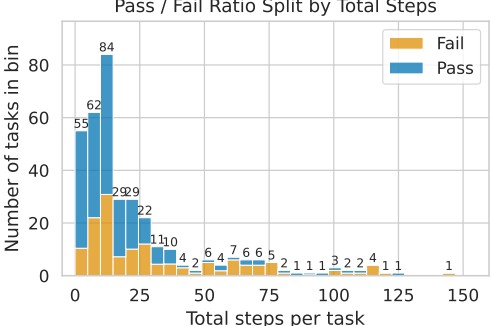

**(c)** Breakdown of passed tasks by application domain, highlighting that coding actions are most frequently applied in complex domains like LibreOffice Calc, Multi-Apps, and direct OS interaction.

**(d)** Pass/fail ratio split by total steps, demonstrating that the failure rate is positively correlated with the number of actions required.

**Figure 3:** CoAct-1 Efficiency and Step Modality Analysis.

budget, rising from 21.4% at 15 steps to 52.5% at 100 steps, highlighting its capacity to solve more complex, long-horizon problems.

In summary, the consistent, state-of-the-art performance across two distinct and challenging benchmarks confirms that CoAct-1's hybrid agentic architecture represents a significant advancement toward creating more capable and reliable autonomous agents for general computer use.

## 4.5 DISCUSSION

**Efficiency analysis** The analysis of CoAct-1's operational efficiency, illustrated in Figure 3, reveals that our hybrid approach is substantially more efficient than leading GUI-only agents. This efficiency is a key factor in its improved success rate. As shown in Figure 3a, CoAct-1 solves tasks with an average of 10.15 steps. This represents a significant improvement over other high-performing agents like GTA-1, which requires 15.22 steps, and UI-TARS, which needs 14.90 steps on average. While OpenAI CUA 4o averages fewer steps (6.14), its overall success rate is much lower compared to CoAct-1's (31.40% v.s. 59.93% on 100 steps). This indicates that CoAct-1's efficiency is coupled with greater effectiveness. The source of this efficiency lies in the strategic use of coding actions. Figure 3b supports this by showing that coding actions help keep the total steps per task relatively low. This efficiency is crucial for robust performance. Figure 3c shows that coding is particularly beneficial in complex domains like "LibreOffice Calc", "Multi-apps", and direct OS interactions, where a large proportion of tasks are solved with code. A single script can replace a long and error-prone sequence of GUI clicks, streamlining the workflow. Figure 3d illustrates a clear trend: tasks that require more actions are more likely to fail. By reducing the total number of steps, the hybrid approach not only accelerates task completion but also minimizes the opportunities for error.

**Table 3:** Performance of CoAct-1 with different backbone model for each participant agent. Powerful Orchestrator significantly help improve the performance on OSWorld.

| GUI Operator | Orchestrator | Programmer | Performance |
|---|---|---|---|
| | o4-mini | o4-mini | 43.43 |
| OpenAI CUA 4o | o3 | o3 | 58.72 |
| | o3 | o4-mini | 60.76 |

**Table 4:** Ablation study on the performance of CoAct-1's components. We compare the full hybrid system against agents restricted to using only the Programmer (pure coding) and only the GUI Operator (pure GUI). The results highlight that the integrated CoAct-1 system significantly outperforms either single-modality agent, demonstrating the effectiveness of its hybrid approach.

| CoAct-1 | | Office | Daily | Professional | OS | Multiple Apps | Avg. | Avg. Steps |
|---|---|---|---|---|---|---|---|---|
| w/ Programmer | w/ GUI Operator | | | | | | | |
| ✓ | | 40.88 | 16.17 | 53.06 | 62.50 | 29.63 | 35.73 | **1.14** |
| | ✓ | 43.50 | 58.80 | 69.38 | **79.16** | 35.68 | 50.68 | 11.20 |
| ✓ | ✓ | **64.80** | **66.51** | **71.82** | 75.00 | **47.87** | **60.76** | 10.15 |

The ability to dynamically select the most appropriate action—either a direct coding command or a GUI interaction—is fundamental to the enhanced efficiency and reliability of CoAct-1.

**CoAct-1 with different backbone**   We investigated the impact of backbone model selection for each agentic component of CoAct-1 on the OSWorld benchmark, with results presented in Table 3. Our analysis reveals that the overall system performance is highly sensitive to the reasoning and instruction-following capabilities of the models chosen for the Orchestrator and Programmer roles. When utilizing o4-mini for both the Orchestrator and Programmer, alongside the OpenAI CUA 4o as the GUI Operator, the system achieved a performance of 43.43%. A significant performance enhancement to 58.72% is observed when a more powerful model, o3, is used for both the Orchestrator and Programmer. This underscores the critical role of a sophisticated high-level planner and a capable code generator in the system's success. The highest performance of 60.76% was achieved with a heterogeneous configuration: employing o3 for the Orchestrator, o4-mini for the Programmer, and retaining OpenAI CUA 4o for the GUI Operator. This configuration suggests an optimal balance, leveraging the powerful reasoning of o3 for task decomposition and delegation, while benefiting from the specialized capabilities of o4-mini for code generation. These results highlight that enhancing the capabilities of the Orchestrator and Programmer yields the most substantial performance gains, validating our modular design and demonstrating the benefits of strategically allocating powerful models to roles with high reasoning demands.

**Pure GUI action VS pure coding action**   To isolate each agent modality and validate our hybrid design, we conducted an ablation study comparing the full CoAct-1 system with agents restricted to a single action type. Results in Table 4 show the combined approach is superior. A Programmer-only agent (pure coding) achieved 35.73% success, highlighting that many tasks require GUI interaction beyond scripting. It was highly efficient, averaging just 1.14 steps per success, confirming the directness of programmatic actions. A GUI Operator–only agent (pure GUI) achieved a higher 50.68% success rate, handling more task types, but required 11.20 steps per task. The full CoAct-1 model, integrating both modalities, achieved 60.76% success with 10.15 steps on average, demonstrating the synergy of our architecture: the Orchestrator exploits the Programmer's efficiency for backend tasks and the GUI Operator's versatility for visual navigation, yielding a robust system.

## 5   CONCLUSIONS

In this work, we introduced CoAct-1, a novel multi-agent system designed to address the inherent inefficiency and brittleness of agents that rely exclusively on GUI manipulation. Our multi-agent system features an Orchestrator that dynamically delegates subtasks to a GUI Operator or a Programmer. Our extensive evaluation on the OSWorld and WindowsAgentArena benchmark confirms the effectiveness of this approach. CoAct-1 achieves a new state-of-the-art success rate of 60.76% on

OSWorld and 52.5% on WindowsAgentArena, significantly outperforming previous leading methods. The performance gains were particularly pronounced in categories involving OS-level interactions, multi-application workflows, and other tasks where the Programmer agent could leverage direct programmatic execution.

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

## A   LLM USAGE STATEMENT

We used a large language model (OpenAI's o3 and GPT-5) as a general-purpose writing assistance tool. Its role was limited to sentence- and paragraph-level polishing, including improving clarity, grammar, and flow. The authors developed all research ideas, analyses, results, and conclusions. The model did not generate new content, perform a literature review, or contribute to the conceptual framing of the paper.

## B   ETHIC STATEMENT

Our research aims to advance computer automation for beneficial and productive purposes, but we acknowledge the potential for dual-use and associated risks.

**Security and Misuse**: An agent with the ability to execute code (Python or Bash) and manipulate a GUI could be leveraged for malicious activities if not properly constrained. To mitigate this risk during our research, all experiments were conducted within secure, isolated, and virtualized benchmark environments (OSWorld and WindowsAgentArena). This ensures that the agent's actions are sandboxed and cannot affect real-world systems or data. We advocate that any future deployment of such agents in live environments must incorporate robust security protocols, strict permission controls, and mechanisms to prevent the execution of harmful code.

**Data Privacy**: The agent's operation relies on observing the screen via screenshots, which in a real-world scenario could contain sensitive or personal information. In this work, we use only the data provided within the benchmark tasks, which do not involve real user data. For any future applications, it is imperative to implement strict data handling policies and privacy-preserving techniques to protect user confidentiality.

## C   REPRODUCIBILITY STATEMENT

Reproduction of CoAct-1 requires an accurate use of specific OpenAI models and prompts. Please review the following details to ensure the accurate performance of our work.

### C.1   MODEL USAGE AND ENVIRONMENT SETTING

**Model Usage**   In this work, we use `o3-2025-04-16` for Orchestrator, `o4-mini-2025-04-16` for Programmer, and `computer-use-preview-2025-03-11` for GUI operator. Any open-sourced model can also be adopted by CoAct-1 if it meets the following requirements:

- **For Orchestrator** The Orchestrator requires multi-modality input (image and text) to process all screenshots from the OS for planning. It also requires a strong reasoning ability across different modalities (See Table 3, reasoning ability will largely affect the final performance).

- **For GUI Operator** The GUI Operator can be replaced with any open-sourced vision language action models (VLA) like UI-TARS (Qin et al., 2025) or OpenCUA (Wang et al., 2025b), or planner-grounder approaches like GTA-1 (Yang et al., 2025) and Agent S2 (Agashe et al., 2025).

CoAct-1 require the GUI Operator to have the following two abilities: (1) instruction following ability out of the grounding task (like when to terminate), and (2) computer use ability, including *clicking*, *dragging*, *typing*, and *hotkeys*. Unfortunately, we have not been allocated enough GPU resources for testing these models' performance on CoAct-1 in this work, and you can expect a reduction in average steps and a performance improvement when switching to a stronger model for the GUI Operator.

• **For Programmer** Language model for Programmer should have a strong ability to write Python and Bash (or Powershell in Windows) for solving computing problems. Note that we didn't provide any API or SDK for programmers; however, this can be an extension to your approach for better performance on app-specific tasks, such as Chrome and Thunderbird.

**Environment** The OS environment is also a significant challenge when evaluating computer-using agents. To reproduce our experiment results, all input screenshots to the Orchestrator should accurately reflect the well-initialized system state. Therefore, we recommend waiting 60 seconds after the VM starts before capturing the OS screenshot to ensure the screenshot includes all necessary information for Orchestrator to plan.

### C.2 PROMPT DESIGN FOR COACT-1

The performance of any modern large language model agent system can be largely affected by its prompt or template design (He et al., 2024). In CoAct-1, despite the rules for general behavior like chain-of-thought and verification, we also design rules based on the model limitations. Specifically, we notice that the GUI Operator has a significant hallucination rate when performing self-checks, while the Programmer's file modifications do not reflect in the OS, so the Orchestrator cannot capture and plan for the next step. We mitigate these limitations in our prompt design by allowing Orchestrator to check the result independently and reload the file modified by the Programmer. We put the prompt used for OSWorld and WindowsAgentArena in Table 5, Table 6, Table 7, Table 8, and Table 9.

## D WHEN&WHY COACT-1 FAILS? INSIGHT FOR FUTURE WORKS

To better understand the capabilities and limitations of CoAct-1, this section analyzes failure cases observed during evaluation. Generally, errors in task completion arise from four primary challenges: high-level, ambiguous queries, reflection errors, and hallucinations.

**High-level query** A high-level query is one where the user's instruction does not directly map to a sequence of actions. Instead, it requires the agent first to infer the user's underlying intent and the broader context before it can devise a solution. For instance, one task in the VSCode domain instructed the agent: "Please help me modify the setting of VSCode to keep my cursor focused on the debug console when debugging in VSCode, instead of automatically focusing back on the Editor." In this scenario, the Orchestrator delegated the task to the Programmer. The Programmer attempted to find the relevant setting by searching for keywords like "debug" and "console". However, it failed to make the conceptual leap that the debugging process relates to "breakpoints." Consequently, it overlooked the correct setting, "focusEditorOnBrake," leading to the task's failure. This case highlights a limitation in the agent's ability to reason about concepts that are not explicitly mentioned in the query.

**Ambiguous query** An ambiguous query is a user request that is vague or omits critical information necessary for successful task completion. Resolving ambiguity often requires the agent to correctly infer the user's intent, which can also involve safety considerations. An example of this occurred in a VSCode task with the instruction: "Please help me modify VSCode setting to hide all "__pycache__" folders in the explorer view." The Orchestrator assigned this subtask to the Programmer. The Programmer successfully identified the need to modify a settings file but incorrectly altered the workspace-specific settings instead of the global user settings. This misinterpretation of the query's scope resulted in the task failing. This illustrates the challenge the agent faces in disambiguating the user's intent when multiple valid interpretations exist.

**Reflection error** Reflection is a crucial mechanism for verifying the task completion process in CoAct-1. In our system design, only the final state of the OS will be returned to the orchestrator after the GUI operator completes the task. This issue yields the final error if the GUI Operator makes some

middle-state errors and the new operation covers them. For example, when performing spreadsheet operations, the GUI Operator may mistype in a cell (let's say a cell in row A), then scroll down the spreadsheet and stop there. In this case, our method will return a screenshot of the OS that excludes the error value in row A. This error cannot be captured by the Orchestrator and will cause the task to fail due to the unexpected operation

**Hallucination** As one of the most important topics in the large language model era, hallucination also appears as a common reasons that cause CoAct-1 failures. All agents in CoAct-1 will hallucinate when task-solving, and the most significant hallucination comes from the Orchestrator and GUI Operator. The Orchestrator will provide an error plan, usually including advanced forecasting, that affects the GUI Operator and Programmer. For example, the Orchestrator may predict the content of an unopened website and instruct the GUI Operator to work on the non-existing content. The GUI Operator will also hallucinate the reasoning process and imagine that it has already completed the assigned task. In this work, we mitigate hallucination from Orchestrator by prompting the Orchestrator to perform verification more frequently and cross-verify the results between the GUI Operator and the Programmer.

---

**System Prompt for Orchestrator (Part 1)**

- - - - - - - - - - - - - - - - - - - - - - - - - - - - - - - - - - - - - - - - - - - - -

Today is {today}.

## Your Role
You are responsible for completing a computer-based task, step by step, using the tools provided.
You are working on a {system_info} system.

### Step-by-Step Process
1. **Describe the Screenshot**
- Carefully review and clearly describe the screenshot's content.

2. **Plan the Task**
- Create a detailed, step-by-step plan to solve the task.
- List all user requirements, including exact file names, file paths, and any other specifics in the output (not in the thinking).

3. **Execute the Instructions**
- Think carefully and follow the user's instructions exactly. **Do not** make any changes not requested by the user (such as renaming files or changing file content).
- You **must** apply all the changes to the computer.
- If the task is impossible (e.g., missing files, wrong environment), reply with **INFEASIBLE** to end the conversation.
- For file operations (like modifying spreadsheets), you MUST try the Programmer (call_programmer) first.
- When the user ask you to create a new sheet in spreadsheet, always name it sequencially. For example, 'Sheet1', 'Sheet2', etc.

4. **Verify the Result**
- **ALWAYS** check the result through the screenshot by yourself. You can let a GUI Operator to navigate to the correct location for you. After the GUI Operator complete, the screenshot will automatically be returned to you.
- If you used the Programmer to modify a file, have the GUI Operator reopen the file to see the updated results.
- Ensure that the result meets all user requirements.
- All the things out of the user's instructions should not be changed.

**Table 5:** System Prompt for Orchestrator (part 1).

**System Prompt for Orchestrator (Part 2)**

- - - - - - - - - - - - - - - - - - - - - - - - - - - - - - - - - - - - - - - - - - - -

## Tools You Can Use

### Programmer (call_programmer)
- Can run Python or Bash code to perform most file or system tasks.
- Needs a clear environment description and detailed task instructions.
- Can use any Python package you specify.
- After modifying a file, ALWAYS verify every change by yourself. You can let a GUI Operator to navigate to the correct location and check the result by yourself. If something is wrong, tell the Programmer to fix it.
Programmer will return a summary of its task solving process after completing the task. No screenshot is provided after the Programmer completes the task.

### GUI Operator (call_gui_operator)
- Can interact with the GUI by clicking on a exact position, scrolling, dragging, typing, and using hotkeys.
- Require a detailed task description.
- The GUI Operator may not able to complete your task in 100% of accuracy and often make mistakes.
- Have a **25-step limit**, each step is a single OS interaction (one click, one hotkey/typing action, etc.).
- **Do not** let the GUI Operator to do any result check. You need to do it by checking the screenshot yourself.
I will return a screenshot that reflect the final state of the computer after completing the task. You don't need to prompt the GUI Operator do this.

**Note:** Only call ONE tool (call_programmer or call_gui_operator) per reply.

**Table 6:** System Prompt for Orchestrator (part 2).

**System Prompt for GUI Operator**

- - - - - - - - - - - - - - - - - - - - - - - - - - - - - - - - - - - - - - - - - - - -

# Your role
You can control the computer by clicking, scrolling, dragging, and typing. Think carefully and execute the user's step-by-step instructions.

# Credentials
The user's password is "{CLIENT_PASSWORD}". Use it when a system password prompt appears.

# Operating rules
- Keep apps open at the end of the task.
- If the UI doesn't appear, perform a brief, deterministic retry (e.g., refocus and re-click).
- Do not close the window, minimize the window unless told to do so.

# Response protocol
When you think the requested task is completed or cannot be completed, reply **exactly**:
'TERMINATE: <1. detailed description of what you see currently. As detailed as possible. 2. What you did to complete the task or why this task cannot be completed.>'

**Table 7:** System Prompt for GUI Operator.

**System Prompt for Programmer**

- - - - - - - - - - - - - - - - - - - - - - - - - - - - - - - - - - - - - - - - - - - -

# Your role
You are the **lead programmer**. Solve the user's task step by step using the terminal (supports Python and Bash).
Your username is 'user'; the sudo password is 'CLIENT_PASSWORD'.
The terminal streams real-time execution output.

# Coding format
Submit **one** fenced code block **only**, labeled with its language:
```bash
# Your Bash script here
# To use sudo, follow this pattern:
echo CLIENT_PASSWORD | sudo -S <your commands>
```

or

```python
# Your Python code here
# Do not use: if __name__ == "__main__": (it will suppress output)
```

# Requirements
- **File names:** Do not rename files or change extensions during any file operation unless the user explicitly asks.
- **Code fence language:** Every fenced block must specify the language ('bash' or 'python'); otherwise you will receive 'unknown language unknown'.
- **Single block:** Wrap all code in **one** code block—do not split your submission across multiple blocks.
- **Spreadsheets:** When editing spreadsheets, ensure **every value** is written to the **intended cell** and **preserve the original formatting** (fonts, colors, sizes, etc.).
- **Dependencies:** Before importing or using a package, **check whether it is installed**; if not, install it in your submission.
- **Observability:** Print intermediate results to aid debugging, for example, the value you are modifying.
- **Final review:** Before completion, carefully inspect your result by writing test cases and confirm that nothing outside the user's instructions has changed.

**Table 8:** System Prompt for Programmer.

**Prompt for LLM Summarizer**

-------------------------------------------------------------------------

# Programmer <-> Terminal Log Summarizer — (No Timeline/Env/Next Actions)

**Role:** Summarize Programmer <-> Terminal logs for the **Orchestrator** so they can decide the next step immediately.
**Orchestrator's task:** '{task}'
**Execution history:** '{chat_history}' (prompts + outputs).

## Output

**1) Summary (2-4 lines)** — task, what was tried, current status, why (cite key log lines / exit codes).

**2) Commands (deduped)**
'''bash
# unique commands in run order; annotate repeats (xN)
'''

**3) Terminal excerpts**
'''text
# minimal evidence: head(~10) ... [truncated N lines] ... tail(~10)
# always include full error traces and return codes
'''

**4) Artifacts / Side effects** — files/dirs changed (paths + purpose); installs/migrations.
*Spreadsheet:* list cells/ranges edited and confirm formatting preserved.

**5) Errors / Blockers** — precise messages + exit codes; likely root cause from logs (no speculation).

**6) Verification** — what checks passed (tests, file existence, row counts); what still needs verification (e.g., reopen file and confirm cell Y).

## Rules
- **Evidence-first**, no speculation.
- **Deterministic truncation** (head/tail; note omitted lines); always include error stacks.
- **Call out deltas** (what changed vs intended).
- **Keep it tight**: bullets > prose.

**Table 9:** Prompt for LLM Summarizer.

