# OpenReview forum: "CoAct-1: Computer-using Multi-agent System with Coding Actions"
_ICLR.cc/2026/Conference — ICLR 2026 Poster_

### Official Review · Reviewer_KLTJ · 2025-10-25

**Soundness:** 2
**Presentation:** 2
**Contribution:** 2
**Rating:** 4
**Confidence:** 3

**Summary:**

This paper presents CoAct-1, a multi-agent system that combines GUI operations with programmatic execution for autonomous computer control tasks. The system comprises three specialized agents: an Orchestrator (for task decomposition and delegation), a Programmer (for writing and executing Python/Bash scripts), and a GUI Operator (for vision-based interface interactions). On the OSWorld and WindowsAgentArena benchmarks, CoAct-1 achieves success rates of 60.8% and 52.5% respectively, significantly outperforming existing methods while reducing average steps from 15 to 10.15.

**Strengths:**

- The paper addresses an important and practical problem regarding the brittleness and inefficiency of pure GUI-based agents on complex, long-horizon tasks. Introducing coding as an augmented action space for computer-using agents is an innovative and promising research direction that offers a new paradigm for general computer automation.

- The multi-agent architecture is well-designed, with the Orchestrator dynamically delegating subtasks to either the Programmer or GUI Operator, effectively combining efficient programmatic execution for backend operations (file management, data processing) with GUI interaction for frontend tasks. The hierarchical memory design (isolated working memory vs. long-term planning memory) effectively prevents context pollution.

- Comprehensive evaluation on two challenging real-world computer operation benchmarks demonstrates significant performance improvements (4.78% over the best baseline on OSWorld, 22.7% on WindowsAgentArena). The efficiency analysis is thorough, clearly showing the advantages of code actions over GUI action sequences, particularly in Calc, VS Code, and multi-application tasks.

**Weaknesses:**

- Imprecise Action Space Definition: The paper defines the action space as A = A_GUI ∪ A_Code, but these two action types have fundamentally different granularities and abstraction levels. A single Python script may be equivalent to hundreds of GUI actions; this simple set union representation obscures the actual complexity and asymmetry.

- Evaluation is limited to two benchmarks, both on Linux/Windows desktop environments. Validation on web applications, mobile platforms, or other operating systems is missing. As to the baseline,  add comparison with Cradle[1] for evaluation.

- Comparisons with baselines are not entirely fair: CoAct-1 uses the latest o3 model while some baselines use 7B open-source models. Comparisons with equivalent-scale models should be provided.

[1] Tan, Weihao, Wentao Zhang, Xinrun Xu, Haochong Xia, Ziluo Ding, Boyu Li, Bohan Zhou et al. "Cradle: Empowering foundation agents towards general computer control." arXiv preprint arXiv:2403.03186 (2024).

**Questions:**

- When the Programmer's code produces errors or unexpected results, how does the system recover? Is there a rollback mechanism? Can the Orchestrator detect Programmer failures and replan?

- What are the API costs？

- Can CoAct-1 generalize to web automation (e.g., Selenium) or mobile application control? What modifications are needed?

---

> ### Author Response · Authors · 2025-11-18
>
> We thank the reviewer for their constructive feedback and insightful questions. We are encouraged that the reviewer found our work flexible and are grateful for the opportunity to clarify these points.
>
> > **[W1] Imprecise Action Space Definition**
> ---
> We thank the reviewer for this insightful point. Our notation $A = A_{\text{GUI}} \cup A_{\text{Code}}$ in `Section 3.1` is intended to formalize the hybrid nature of the action space available to the overall system, not to imply a flat, atomic action space for a single policy. The core of our contribution is managing this very asymmetry through our hierarchical, multi-agent architecture. As detailed in `Section 3.2`, the high-level Orchestrator ($\pi_{\text{orch}}$) acts as a meta-policy that makes a strategic choice between these modalities: delegating to either the GUI Operator ($\pi_{\text{GUI}}$) or the Programmer ($\pi_{\text{Code}}$). Each executor agent then engages in its own complex, multi-round interaction (e.g., a "perception-action" loop for the GUI Operator or a "coding-observation" circle for the Programmer) to solve the subtask. Thus, the union represents the high-level delegation choice for the Orchestrator, while the system's hierarchical structure is explicitly designed to handle the different levels of abstraction.
>
> > **[W2] Imprecise Action Space Definition**
> ---
> We appreciate the reviewer's suggestions for broadening the evaluation. We selected OSWorld and WindowsAgentArena as they are the most challenging and comprehensive benchmarks currently available for general-purpose desktop automation on real operating systems, which is the primary focus of our work and most of the other baselines `[1, 2, 3, 4]`. These benchmarks test long-horizon planning and interaction with heterogeneous GUI environments. **We would like to clarify that our evaluation already includes web application tasks; specifically, the OSWorld "Daily tasks" category includes tasks in Chrome , and WindowsAgentArena features a dedicated "Web" category with tasks in Chrome and Microsoft Edge.** Our strong performance on these benchmarks therefore already provides validation on web applications as part of a general desktop workflow. We believe demonstrating state-of-the-art performance on these complex, multi-application desktop benchmarks is a significant contribution.
>
> We also put the comparison between CoAct-1 and Cradle in the following table:
> | **Methods**                      | **Performance on OSWorld** |
> |----------------------------------|----------------------------|
> | Cradle (w/ GPT-4o, in the paper) | 7.8                        |
> | Cradle (w/ o3)                   | 13.7                       |
> | CoAct-1 (w/ o3)                  | **60.8**                       |
>
> Obviously, CoAct-1 performs better than Cradle on OSWorld.
>
> Refs:
>
> [1] Agashe, Saaket, et al. "Agent s2: A compositional generalist-specialist framework for computer use agents." COLM (2025).
>
> [2] Guo, Liangxuan, et al. "Agentic Lybic: Multi-Agent Execution System with Tiered Reasoning and Orchestration." arXiv preprint arXiv:2509.11067 (2025).
>
> [3] Fu, Tianyu, et al. "Mano Technical Report." arXiv preprint arXiv:2509.17336 (2025).
>
> [4] Agashe, Saaket, et al. "Agent s: An open agentic framework that uses computers like a human." ICLR (2024).
>
> > **[W3] Comparisons with baselines are not entirely fair**
> ---
> We thank the reviewer for raising this important point on model equivalence. We want to clarify that our primary and most critical comparisons are indeed with baselines using equivalent-scale models. As shown in `Table 1`, **all of our agentic baselines use the powerful OpenAI o3 model as their high-level planner with an equivalent or even powerful GUI Agent (UI-TARS-1.5 (7B), GTA-1, etc)**. However, our system (60.76% on OSWorld) still significantly outperforms these baselines.
>
> We also includes results with 7B-scale GUI operator or more powerful GUI operator like Claude-4-sonnat in the following table:
> | Method        | Performance on OSWorld (100 steps) |
> |------------------------------|------------------------------------|
> | UI-TARS-1.5 (7B) [1]         | 27.4                               |
> | Claude-4-sonnat [2]          | 41.4                               |
> | **w/ CoAct-1 (as GUI Operator)** |                                    |
> | UI-TARS-1.5 (7B) [1]         | **60.2 (+32.8)**                       |
> | Claude-4-sonnat [2]          | **64.2 (+22.8)**                       |

---

> ### Author Response · Authors · 2025-11-18
>
> > **[Q1] When the Programmer's code produces errors or unexpected results, how does the system recover? Is there a rollback mechanism? Can the Orchestrator detect Programmer failures and replan?**
> ---
> This is a critical question about system robustness. Error recovery is managed at two levels, as described in our multi-agent design (`Section 3.2`) and prompt design (`Appendix C.2`). **First, the Programmer agent itself operates in a multi-round coding-observation loop**; it receives direct feedback (e.g., stack traces, print outputs) from the code interpreter and can reflect and refine its code to fix syntax errors or runtime failures. **Second, for semantic errors where the code runs but produces an incorrect outcome, recovery is handled by the Orchestrator.** Upon subtask completion, the Orchestrator receives a summary and a new observation. As detailed in our prompt design (`Table 5`, `Table 6`), the Orchestrator is explicitly instructed to **independently verify** the outcome. For example, after the Programmer modifies a file, the Orchestrator will delegate a new task to the GUI Operator to "reopen the file to see the updated results". If this verification fails, the Orchestrator detects the failure, replans, and can re-delegate to the Programmer with corrective instructions. While we do not currently implement an explicit state-rollback mechanism, this hierarchical verification and replanning loop allows the system to correct failures.
>
> > **[Q2] What are the API costs？**
> ---
> The API cost for complete the 361 tasks on OSWorld is as follows:
> | **Model**                            | **Cost (US Dollar)** |
> |--------------------------------------|----------------------|
> | OpenAI o3                            | $71.14               |
> | OpenAI o4-mini                       | $34.27               |
> | computer-use-preview (OpenAI CUA 4o) | $870.48              |
>
> The main cost comes from the GUI Agent as it needs to take multiple high-resolution images as input in each call. One way to reduce the cost is to switch the computer-use-preview into open source models like UI-TARS-1.5 (7B).
>
> > **[Q3] Can CoAct-1 generalize to web automation (e.g., Selenium) or mobile application control? What modifications are needed?**
> ---
> This is an excellent question about the generalizability of our hybrid paradigm. We believe the core architecture of CoAct-1 is **highly adaptable** to other domains like web and mobile automation. The primary modification would be swapping the execution environments for the specialized agents. For web automation, the GUI Operator (a VLM) could remain, or be replaced by a DOM-based agent. More importantly, the Programmer agent, which currently executes Python and Bash, could be empowered to write and execute scripts in a different context, such as generating Python scripts using the **Selenium** library or injecting direct **JavaScript** snippets. For mobile control, the Programmer could similarly be adapted to generate and execute `adb` shell commands or scripts for mobile-specific frameworks (e.g., `Appium`). The fundamental principle of bypassing brittle, multi-step GUI sequences with robust, single-shot code execution remains the same. The main adaptation is thus to equip the Programmer with the appropriate SDK or interpreter for the target platform, which is a promising direction for future work.

---

> > ### Comment · Reviewer_KLTJ · 2025-11-22
> >
> > Thank you for your detailed response. Your clarifications have fully addressed my concerns, so I am raising my score accordingly.

---

### Official Review · Reviewer_T8jB · 2025-10-25

**Soundness:** 4
**Presentation:** 4
**Contribution:** 2
**Rating:** 4
**Confidence:** 4

**Summary:**

This paper presents CoAct-1, a multi-agent framework to tackle long-horizon computer-use tasks. By combining an orchestrator, a GUI agent, and a programming agent, this work achieves state-of-the-art results on OSWorld and WindowsAgentArena.

**Strengths:**

1. Good engineering results and practical value. This work achieved SOTA (at the time of ICLR submission) on two OS use benchmarks, covering both Linux and Windows environments.

2. Clear methodology and good writing. Ablation studies are well-designed and insightful. Full prompts, model versions, and detailed implementation notes are provided, demonstrating a high level of transparency.

**Weaknesses:**

1. Limited research contribution. The paper reads more like a well-written technical report. The approach primarily combines existing and widely-used components (coding, visual, a multi-agent orchestrator) without proposing fundamental methodological advances. It has limited differentiation from general tool-use agents.

2. The OSWorld task distribution is skewed towards Office tasks, specifically, LibreOffice Calc and LibreOffice Writer, where programming agents could greatly outperform pure visual agents. It is unclear if this approach would be particularly beneficial to other daily tasks, and if so, why.

3. Lack of experiments on different models. This work primarily uses OpenAI CUA 40 as the GUI operator and has only explored o3 and o4-mini as the orchestrator and programmer.

**Questions:**

1. CoAct-1 doesn't perform particularly well with a very limited step budget (15 steps), while this work has expressed that the use of a coding agent saves steps. How would you explain this contradiction?

2. In Figure 1, why does the Programmer agent return a screenshot as a summary?

---

> ### Author Response · Authors · 2025-11-18
>
> We thank the reviewer for their constructive feedback and insightful questions. We are encouraged that the reviewer found our work flexible and are grateful for the opportunity to clarify these points.
>
> > **[W1] Limited research contribution.**
> ---
> We respectfully disagree with the limited differentiation from general tool-use agents assessment. Tool-use agents typically call predefined, discrete APIs. **Our contribution is the conception and implementation of a new, more powerful paradigm: free-form coding as a first-class, dynamic action modality** that co-exists with GUI actions for computer-using tasks. This hybrid, dual-modality design is a fundamental methodological advance that our paper is the first to propose and empirically validate as a state-of-the-art approach.
>
> On the other hand, we are **not just simply combining components**. A naive "plug-and-play" combination is insufficient for complex, long-horizon computer tasks, which often fail due to compounding errors and unmanageable context. Our key methodological contribution is the hierarchical workflow with isolated executor memories, as detailed in `Section 3.3`: (1) the Orchestrator maintains the long-term, high-level plan, and (2) the GUI Operator and Programmer use ephemeral, short-term working memories that are cleared after each subtask is completed. This deliberate memory isolation is our novel solution to the long-context challenge. It is the "how" that makes our system work.
>
> > **[W2] It is unclear if this approach would be particularly beneficial to other daily tasks, and if so, why.**
> ---
> We thank the reviewer for this insightful comment regarding the task distribution. While OSWorld is indeed weighted towards Office tasks, we respectfully disagree that our method's benefits are confined to this domain; in fact, our results demonstrate the broad and significant advantages of our hybrid approach across all task categories. As shown in `Table 1`, our agent achieves a `66.51%` success rate on the 78 Daily tasks (Chrome, Thunderbird, VLC), which is notably higher than its `64.80%` rate on the 104 Office tasks. Furthermore, our agent achieves its highest performance in OS (`75.00%`) and Professional (`71.82%`) tasks, underscoring its general applicability beyond Office. This trend is further confirmed on the WindowsAgentArena benchmark (`Table 2`), where our method excels in non-Office categories such as Windows System (`83.3%`) and Windows Utils (`77.7%`).
>
> **The core benefit of our approach stems not from an over-reliance on programming, but from the Orchestrator's ability to dynamically delegate to the best-suited agent.** This synergy is the key: our agent strategically uses code for efficient backend operations (e.g., file management, data processing) common in OS or Multi-Apps tasks, while seamlessly reverting to robust GUI manipulation for visually-driven "Daily" tasks (as shown in `Fig. 3c`), thus proving its value across the entire spectrum of computer-use problems.
>
> > **[W3] Lack of experiments on different models.**
> ---
> We want to clarify that the signal we want to give for the reader is two-fold: (1) Orchestrator’s planning ability can significantly affect the final performance, and (2) a stronger Orchestrator and GUI operator can lead to better performance. Our experiments have clearly delivered the first signal as we test CoAct-1 with the orchestrator in different scales (`Table 3`). For the second part, please see the following table (with o3 as orchestrator and o4-mini as Programmer):
> | Method        | Performance on OSWorld (100 steps) |
> |------------------------------|------------------------------------|
> | UI-TARS-1.5 (7B) [1]         | 27.4                               |
> | Claude-4-sonnat [2]          | 41.4                               |
> | **w/ CoAct-1 (as GUI Operator)** |                                    |
> | UI-TARS-1.5 (7B) [1]         | **60.2 (+32.8)**                       |
> | Claude-4-sonnat [2]          | **64.2 (+22.8)**                       |
>
> Please also note that we use the weakest close-source computer-using agent (OpenAI CUA 4o) for CoAct-1 as a fair comparison with other methods. Switching to other stronger CUA can lead to significantly better results.
>
> Refs:
>
> [1] Qin, Yujia, et al. "Ui-tars: Pioneering automated gui interaction with native agents." arXiv preprint arXiv:2501.12326 (2025).
>
> [2] Antrophic. “System Card: Claude Opus 4 & Claude Sonnet 4.” (2025).

---

> ### Author Response · Authors · 2025-11-18
>
> > **[Q1] CoAct-1 doesn't perform particularly well with a very limited step budget (15 steps), while this work has expressed that the use of a coding agent saves steps. How would you explain this contradiction?**
> ---
> We want to clarify that **we achieve the second-best performance at 15 steps**. The Jedi-7B, which outperformed CoAct-1 at 15 steps, leveraged many handcrafted tools to achieve better performance, whereas we didn’t. In CoAct-1, we only provide necessary instructions, no few-shot examples, and no hand-crafted tools. Those hand-crafted tools can be treated as a pre-processed summarization for many necessary early-experience steps, which derives the reason of why CoAct-1 performs worse at 15 steps than Jedi-7B.
>
> > **[Q2] In Figure 1, why does the Programmer agent return a screenshot as a summary?**
> ---
> We regret that this was a mistake. The summary was not provided as a screenshot by the Programmer. Instead, the summary was generated by an LLM summarizer by looking into the interaction trajectory between the Programmer and the OS.

---

> > ### Comment · Reviewer_T8jB · 2025-11-19
> >
> > Thanks for the explanation, which clears some of my concern. I have raised the rating accordingly.

---

### Official Review · Reviewer_x4pG · 2025-10-28

**Soundness:** 4
**Presentation:** 4
**Contribution:** 4
**Rating:** 8
**Confidence:** 4

**Summary:**

The paper introduces of CoAct-1, a multi-agent system that combines both a programmer agent and GUI agent. The system contains the following module:

An Orchestrator, which serves as a high-level planner to orchestrate tasks.
A GUI Operator, a standard VLM agent that performs visual actions like moving mouse, and using keyboard.
A Programmer, a specialized agent that can write and execute Python or Bash scripts to perform backend operations like file management or data processing.

The main idea is that the Orchestrator can dynamically delegate a subtask to the most efficient agent, leveraging the advantage of both GUI agent and programming agent.
The authors evaluate CoAct-1 on the OSWorld and WindowsAgentArena benchmarks. They show the state of the art performance and a improved efficiency.

1. CoAct-1 achieves a new SOTA success rate of 60.8% on OSWorld and 52.5% on WindowsAgentArena, significantly outperforming prior GUI-only methods.

2. The hybrid approach reduces the average number of steps required for a successful task on OSWorld to 10.15, compared to ~15 steps for leading GUI-only agents.

**Strengths:**

The paper is well written. The idea of a hybrid computer use agent is novel. The experiment is solid and covers different OS platforms. The overall presentation is good. The authors are also able to demonstrate the success rate increase and steps reduced from this CoAct-1 clearly over the current SoTA agents in the OSWorld benchmark and the WindowsAgentArena benchmark. The authors provide a detailed example of a user task and provide the detailed prompt of each module, which is useful for the reader to enhance understanding. This work meets the quality requirements for this conference.

**Weaknesses:**

Overall this is solid work and I don't have any specific questions or concerns.

**Questions:**

see above

---

> ### Author Response · Authors · 2025-11-18
>
> We sincerely thank the reviewer for their exceptionally positive feedback and strong support for our paper. We are delighted that the reviewer found our hybrid agent idea novel and the paper well written and solid. We especially appreciate their recognition of our solid experimental methodology and that we clearly demonstrate CoAct-1's state-of-the-art success and efficiency gains on both the OSWorld and WindowsAgentArena benchmarks. As the reviewer noted they have no specific questions or concerns, we simply wish to express our gratitude for their thorough and supportive assessment.

---

> > ### Comment · Reviewer_x4pG · 2025-11-25
> >
> > Thanks for the authors. I have no other concerns.

---

### Official Review · Reviewer_PXQ7 · 2025-11-08

**Soundness:** 3
**Presentation:** 3
**Contribution:** 2
**Rating:** 6
**Confidence:** 2

**Summary:**

CoAct-1 proposes a hybrid approach for autonomous computer agents that leverages both GUI manipulation and programmatic scripting to execute tasks more efficiently and robustly. The system introduces three specialized agents—Orchestrator, Programmer, and GUI Operator—to dynamically switch between GUI actions and Python/Bash scripting based on task requirements. The approach is evaluated on benchmarks like OSWorld and WindowsAgentArena, demonstrating significant performance improvements in success rates and task completion efficiency over traditional GUI-only systems.

**Strengths:**

Hybrid Approach: The combination of GUI manipulation and coding as an action creates a powerful, adaptable agent system. The Orchestrator’s dynamic task delegation helps maximize the efficiency of both visual interactions and direct system manipulation.

State-of-the-Art Performance: CoAct-1 sets new benchmarks for both OSWorld (60.8% success rate) and WindowsAgentArena (52.5% success rate), outperforming existing methods like Agent S2.5 and GTA-1 by significant margins. The system excels in tasks requiring complex file management, data processing, and cross-application workflows.

Flexibility: By incorporating both GUI and code-based actions, CoAct-1 can handle a broad range of tasks with varying complexities, from simple GUI interactions to complex backend system operations. This flexibility is especially beneficial for applications requiring precise or multi-round task execution.

**Weaknesses:**

Complexity: The reliance on three distinct agents (Orchestrator, Programmer, and GUI Operator) introduces significant system complexity. While the multi-agent framework allows for high flexibility, it also makes the system harder to manage and debug, especially in real-world applications where the agents may not always coordinate perfectly.

Additionally, I believe that a more in-depth discussion about the differences between this work and other hybrid frameworks would be beneficial.

**Questions:**

How does CoAct-1 handle tasks that require multiple rounds of interaction with both the GUI and programmatic code? Is there a risk of inefficient back-and-forth between agents?

What improvements are planned for enhancing the system’s robustness in handling ambiguous or complex user instructions, especially in non-structured environments?

---

> ### Author Response · Authors · 2025-11-18
>
> We thank the reviewer for their constructive feedback and insightful questions. We are encouraged that the reviewer found our work flexible and are grateful for the opportunity to clarify these points.
>
> > **[W1] On System Complexity**
> ---
> We respectfully disagree that the system's modularity is a weakness; rather, it is its core strength and a deliberate design choice to manage the inherent complexity of general computer automation. **The CoAct-1 system uses a multi-agent design, comprising an Orchestrator for high-level planning, a GUI Operator for visual interaction, and a Programmer for code execution, to overcome the complexity of a single monolithic agent, achieving specialization and robustness.** The Orchestrator manages coordination in a strictly hierarchical workflow, and executors maintain isolated, short-term memories to prevent complex dependencies. An ablation study confirms the efficacy of this design, showing that the full CoAct-1 system (60.76% success) significantly outperforms both the GUI Operator-only (50.68%) and Programmer-only (35.73%) agents, demonstrating that the multi-agent framework provides the necessary synergy to solve tasks single-modality agents cannot.
>
> > **[W2] On Comparison to Other Hybrid Frameworks**
> ---
> We appreciate the suggestion to deepen this comparison. We did situate our work relative to other hybrid agentic frameworks in `Section 2`. We explicitly cite systems like UFO-2, PyVision, BeyondBrowsing, and ALITA , noting they "share the principle of dynamically constructing and invoking tools". The key differentiator of CoAct-1 is its **novel focus on integrating free-form coding as a first-class, enhanced action alongside traditional GUI actions** for general computer operation. While other frameworks compose predefined tools or APIs, CoAct-1's Orchestrator dynamically arbitrates between two distinct modalities of execution (GUI vs. code) to solve a single task. This hybrid, hierarchical policy allows it to strategically bypass brittle GUI sequences (e.g., file manipulation) with robust code, leading to the SOTA efficiency and success rates we report.
>
> > **[Q1] How does CoAct-1 handle tasks that require multiple rounds of interaction with both the GUI and programmatic code? Is there a risk of inefficient back-and-forth between agents?**
> ---
> This is an excellent question that highlights the core function of our Orchestrator. The workflow illustrated in `Figure 2` provides an  example:
> - **User Task:** "unzip the downloaded extension file... and configure it in Chrome's extensions". This task requires both coding (unzip) and GUI (configure).
> - **Round 1 (Code):** The Orchestrator first identifies the file operation and delegates it to the Programmer ("call_coding_agent") with the subtask "Unzip the file...". The Programmer executes a Bash script to do this in a nested conversation with multiple rounds.
>     - **Handoff:** The Programmer completes its task and returns a concise summary to the Orchestrator.
> - **Round 2 (GUI):** The Orchestrator receives this summary, updates its plan, and now delegates the next subtask ("Configure the Unpacked Extension in Chrome") to the GUI Operator ("call_gui_agent"). A GUI Agent will handle this subtask in a nested conversation with multiple rounds.
>     - **Handoff:** The GUI Operator performs the visual navigation (clicking, etc.) and, upon completion, returns its own summary and the final screenshot to the Orchestrator.
> - **(Omitted Rounds)**
> - **Termination:** The Orchestrator reviews the final state and summary, confirms the task is complete, and terminates.
>
> This is not an inefficient "back-and-forth" but a structured, hierarchical delegation. The Orchestrator remains the central planner, making a new, informed decision after each subtask is completed, based on the latest system state (screenshot) and summary.
>
> > **[Q2] What improvements are planned for enhancing the system’s robustness in handling ambiguous or complex user instructions, especially in non-structured environments?**
> ---
> Handling ambiguous user intent is a well-documented challenge. Existing strategies often involve rewriting the intent into a detailed, step-by-step plan for the LLM [1] or applying reinforcement learning in a simulated environment to improve the on-policy model's comprehension [2].
>
> Building on this, our future work will involve constructing a user-interaction computer-use environment. We will then utilize reinforcement finetuning (RFT) to specifically enhance the LLM's ability to interpret ambiguous and high-level queries, as further detailed in `Appendix D`.
>
> Refs:
>
> [1] Qian, Cheng, et al. "Userbench: An interactive gym environment for user-centric agents." arXiv preprint arXiv:2507.22034 (2025).
>
> [2] Qian, Cheng, et al. "UserRL: Training Interactive User-Centric Agent via Reinforcement Learning." arXiv preprint arXiv:2509.19736 (2025).

---

### Author Response · Authors · 2025-11-28
**Note on Rebuttal Updates regarding Review Rollback**

Dear AC,

We sincerely appreciate your efforts in handling this challenging situation.

Regarding the recent data leak and the decision to rollback reviews:

We would like to highlight that prior to this rollback, our rebuttal successfully addressed `Reviewer T8jB`'s concerns regarding research novelty and generalizability, as well as `Reviewer KLTJ`'s questions on action space definition and baseline fairness. **Both reviewers explicitly commented that they had raised their scores to 6 following our response before Nov 27.**

We kindly hope that you consider these positive outcomes and the consensus reached during the discussion when making your final recommendation.

Best regards,

Authors of CoAct-1

---

### Meta-Review · Area_Chair_35Xd · 2026-01-18

**Summary:**

Reviewers largely agreed that the paper addresses an important and practical problem in long-horizon computer-use agents and demonstrates strong empirical performance. The main concerns focused on the novelty of the contribution relative to existing tool-using or hybrid agent frameworks, the complexity of the multi-agent design, fairness of baseline comparisons (model scale and task distribution), and the clarity of the action-space formulation. Some reviewers also questioned generalization beyond OSWorld-style desktop tasks and raised issues around step-budget performance, error recovery, and system cost.

**Reviewer Concerns:**

Several concerns were convincingly addressed in the rebuttal. The authors clarified the technical contribution beyond a naive combination of components, emphasizing free-form coding as a first-class action and the hierarchical orchestration with isolated executor memories, which directly responds to novelty and system-design critiques. Questions regarding coordination between GUI and coding agents, error recovery, and replanning were answered with concrete workflow explanations and verification mechanisms. Concerns about task skew toward Office applications were addressed with category-wise breakdowns showing strong performance on non-Office tasks across both OSWorld and WindowsAgentArena. Baseline fairness issues were mitigated by additional comparisons using matched or weaker GUI operators, as well as explicit comparisons to Cradle. Multiple reviewers explicitly acknowledged that their concerns were resolved and raised their scores accordingly.

Some concerns remain partially open. The conceptual novelty may still feel incremental to reviewers seeking fundamentally new agent-learning paradigms rather than strong system design and integration. Evaluation remains limited to desktop OS environments, with broader validation on mobile or non-desktop settings left as future work. Cost remains relatively high due to GUI agent usage, although this is transparently reported and discussed.

**Reviewer Scores:**

- Reviewer x4pG (initial: 8) remained strongly positive throughout and explicitly stated having no remaining concerns.
- Reviewer PXQ7 (initial: 6) would likely remain around the same score; their main reservations concerned system complexity rather than correctness, and these were addressed but not central to their evaluation.
- Reviewer T8jB (initial: 4) explicitly acknowledged after the rebuttal that their concerns were clarified and stated that they raised their rating accordingly prior to the review rollback.
- Reviewer KLTJ (initial: 4) also explicitly commented that their concerns were fully addressed and that they raised their score following the rebuttal, again prior to the rollback.

---

### Decision · Program_Chairs · 2026-01-26

Accept (Poster)